# XYZ-Text2SQL-R1: Simple Rewards, Strong Reasoning in Text-to-SQL

## Abstract

Translating natural language into SQL (Text2SQL) is a longstanding challenge at the intersection of natural language understanding and structured data access. While large language models (LLMs) have significantly improved fluency in SQL generation, producing correct and executable SQL, particularly for complex queries, remains a bottleneck. We present **XYZ-Text2SQL-R1**, a reinforcement learning (RL) framework and model family designed to generate accurate, executable SQL using a lightweight reward signal based solely on execution correctness. Our approach avoids brittle intermediate supervision and complex reward shaping, promoting stable training and alignment with the end task. Combined with carefully curated data, strong supervised initialization, and effective training practices, XYZ-Text2SQL-R1 achieves state-of-the-art execution accuracy across six diverse Text2SQL benchmarks and ranks among the leading entries on the BIRD leaderboard. Notably, our 7B model outperforms prior 70B-class systems, highlighting the framework's scalability and efficiency. We further demonstrate inference-time robustness through simple extensions like value retrieval and majority voting. Extensive experiments and ablation studies offer both positive and negative insights, providing practical guidance for future Text2SQL research.

## 1 Introduction

Translating natural language questions into SQL queries, commonly known as **Text2SQL**, is a core challenge in natural language understanding and human-computer interaction. A robust solution would enable non-technical users to query structured databases using natural language, thereby democratizing access to data analytics.

Recent advances in large language models (LLMs) have significantly improved the fluency and surface-level coverage of Text2SQL systems Ma et al. (2025); Pourreza et al. (2025); Zhai et al. (2025); Papicchio et al. (2025). However, generating *correct and executable* SQL, especially for complex queries involving multi-table joins, nested logic, and nuanced schema understanding, remains difficult. Most existing approaches rely on supervised fine-tuning over (question, SQL) pairs Li et al. (2025); Gao et al. (2024), which often fail to promote the intermediate reasoning steps essential for reliable and generalizable SQL generation.

We introduce **XYZ-Text2SQL-R1**, a reinforcement learning (RL) framework and model family for generating high-quality executable SQL from natural language. It uses a lightweight reward based solely on execution correctness, avoiding brittle partial rewards while promoting stable training and alignment with the end goal. Combined with best practices in data filtering, synthetic generation, and model-based selection, XYZ-Text2SQL-R1 demonstrates that high-quality data, strong supervised initialization, and a simple reward signal together yield accurate, scalable, and robust Text2SQL models.

**Our main contributions are:**

- **Simple and Scalable RL for Text2SQL:** We develop a lightweight RL formulation using execution-only rewards that generalizes well across model sizes and benchmarks, enabling stable training and strong performance without complex reward design.
- **Comprehensive Benchmark Leadership:** Across six diverse Text2SQL benchmarks, XYZ-Text2SQL-R1 consistently outperforms both SQL-specialized and general-purpose LLMs, including GPT-4o OpenAI (2023) and DeepSeek-V3 Liu et al. (2024), demonstrating robust generalization

across domains. On the BIRD leaderboard, our models are positioned among the leading entries: the 32B model achieves 71.83% execution accuracy, which further improves to 73.84% when applying self-consistency. Among models with publicly available details, XYZ-Text2SQL-R1 ranks first, ensuring transparency, reproducibility, and accessibility. Notably, our 7B model even surpasses prior 70B-class systems Zhai et al. (2025), highlighting the scalability and efficiency of our approach.

- **Data and Training Strategies That Matter:** We present best practices for data filtering, synthetic data generation, and model-based selection. Additionally, we demonstrate the value of *online RL training*, which leverages strong supervised checkpoints and well-designed prompts to further improve performance. Combined with high-quality data and a simple execution-driven reward, these practices produce accurate, scalable, and robust Text2SQL models.
- **Broad and Rigorous Evaluation:** We evaluate XYZ-Text2SQL-R1 across six Text2SQL datasets, covering a range of schema complexity and query difficulty. This provides strong evidence of generalization and guards against overfitting to any single dataset or evaluation format.
- **Inference-Time Robustness and Extensibility:** XYZ-Text2SQL-R1 supports simple inference-time techniques, such as value retrieval and majority voting, that further improve accuracy with minimal system overhead, demonstrating its practicality for real-world deployment.
- **Empirical Insights for the Community:** We report both successful and negative findings, offering transparent and actionable insights to guide future RL-based Text2SQL research/development.

## 2 RELATED WORK

The research presented in XYZ-Text2SQL-R1 draws inspiration from and aims to advance two key areas of investigation: the development of robust Text2SQL systems and the refinement of reinforcement learning techniques for sophisticated language model reasoning.

**Reinforcement Learning for LLM Reasoning.** Recent research has demonstrated the potential of RL techniques to significantly enhance the reasoning capabilities of LLMs OpenAI et al. (2025); Lambert et al. (2024); OpenAI et al. (2024); Guo et al. (2019). By providing explicit rewards for logical correctness and step-wise reasoning, RL enables models to autonomously discover effective problem-solving strategies, often mirroring structured human reasoning Xu et al. (2025); Wang et al. (2025); Yang et al. (2025b). Applications span mathematical problem solving (where RL fine-tuning on step-by-step correctness or final answer accuracy yields substantial improvements Shao et al. (2024); OpenAI et al. (2024)) and code generation, where preference optimization and RL from human feedback have led to greater code validity and efficiency Wang et al. (2025); Yang et al. (2025b).

Most prior methods are built on top of policy gradient algorithms such as Proximal Policy Optimization (PPO) Schulman et al. (2017) or, more recently, Group Relative Policy Optimization (GRPO) Guo et al. (2025); Yang et al. (2024b); Dang & Ngo (2025), which compares groups of generated responses rather than evaluating them in isolation. This approach is particularly powerful for reasoning tasks with multiple plausible solutions, enabling the model to build a deeper understanding of what constitutes high-quality reasoning Kumar et al. (2025). For example, GRPO has enabled models like DeepSeek-R1-Zero to develop complex reasoning skills such as multi-step chain-of-thought simply by being rewarded for correct final answers Guo et al. (2025). The effectiveness of these RL frameworks often hinges on carefully designed preference datasets and reward models that accurately reflect the subtleties of logical reasoning. Our work draws on these insights, by adapting GRPO for the Text2SQL domain and focusing the reward signal on final executable correctness, encouraging the model to reason through the full compositional structure of the query.

**Text-to-SQL.** The Text2SQL task has seen remarkable progress over the years Zhu et al. (2024b). Early systems were built on rule-based approaches and handcrafted grammars, but proved brittle when faced with linguistic ambiguity or complex schema variation Androutsopoulos et al. (1995); Li & Jagadish (2014). The advent of neural sequence-to-sequence models helped automate parts of the semantic parsing process, though such models often required elaborate schema encoders and strong supervision Guo et al. (2019); Wang et al. (2021).

More recently, the generalization power of LLMs has revolutionized the field, as demonstrated by numerous works leveraging pre-trained LLMs for template-free SQL generation Singh et al. (2025); Gao et al. (2023); Li et al. (2024). LLMs often exhibit basic SQL competence on "out-of-the-box" evaluation, but their performance on complex, multi-table queries remains limited unless explicit

compositional reasoning is encouraged Wei et al. (2023); Shao et al. (2024); Guo et al. (2025); OpenAI et al. (2024). This has motivated the development of techniques such as Chain-of-Thought prompting Tai et al. (2023), query decomposition Eyal et al. (2023), optimization Zhai et al. (2025), as well as dynamic schema linking and execution-based feedback Hong et al. (2024); Deng et al. (2025).

Within this landscape, there is growing interest in RL-based approaches for Text2SQL, often building on structured, multi-component reward functions that aggregate execution feedback, syntactic validity, partial string overlap, schema conformance, and more Pourreza et al. (2025); Ma et al. (2025); Papicchio et al. (2025). However, such complex reward engineering risks encouraging superficial reward hacking, as ob-

Table 1: The reward design comparison of XYZ-Text2SQL-R1 to existing reasoning models.

| Framework | Optimization (Reward) |
| --- | --- |
| Reasoning-SQL Pourreza et al. (2025) | GRPO (EX, syntax, n-gram, LLM, schema, format) |
| SQL-R1 Ma et al. (2025) | GRPO (EX, length, syntax, format) |
| Think2SQL Papicchio et al. (2025) | GRPO (precision, recall, cardinality, 2×format) |
| ExCoT Zhai et al. (2025) | DPO (EX) |
| XYZ-Text2SQL-R1 | GRPO (EX, syntax) |

served in RL for semantic parsing Skalse et al. (2022). In contrast, our work advocates for a simpler, execution-centric reward design, inspired by earlier RL-based semantic parsing approaches such as Seq2SQL Zhong et al. (2017), and demonstrates that this minimal reward scheme can be both more stable and more effective in practice.

Compared to most recent reasoning works Pourreza et al. (2025); Ma et al. (2025); He et al. (2025), or pairwise preference optimization Zhai et al. (2025), our method is unique in unifying a streamlined data filtering pipeline, group-based relative policy optimization, and a strictly execution-based reward (Table 1).

## 3 METHODOLOGY

Building on insights from prior RL work for language model reasoning, we design XYZ-Text2SQL-R1 to pursue robust Text2SQL generation via a simple, execution-grounded RL framework.

**Overview of RL Approach.** We adopt GRPO Shao et al. (2024) as it has demonstrated superior efficiency and effectiveness on structured reasoning tasks, making it an ideal match for Text2SQL. Applying RL in this setting is particularly necessary. SQL correctness can be directly verified against databases, providing clear and automatic reward signals. In addition, the generation process inherently involves multi-step decision making. Finally, given the scarcity of annotated data, RL allows the model to explore and generalize to novel compositional patterns beyond the training set.

Formally, let $\pi_\theta$ denote our policy model parameterized by $\theta$. For each input text question $Q$ with associated database schema, the model generates $N$ candidate SQL queries (aka rollouts), $\{o_{Q,1}, ..., o_{Q,N}\}$. Each generated query is then evaluated to provide an explicit reward signal as described in the later of the section. These per-group rollouts allow us to compute relative advantages, stabilizing learning and promoting robust policy improvement.

The GRPO objective is as follows:

$$\mathcal{J}_{\text{GRPO}}(\theta) = \mathbb{E}\left[\frac{1}{N}\sum_{i=1}^{N}\min\left(r_i A_i,\ \text{clip}(r_i, 1-\epsilon, 1+\epsilon)A_i\right)\right] - \beta \mathrm{D}_{\text{KL}}(\pi_\theta \| \pi_{ref}),$$

where $r_i = \frac{\pi_\theta(o_i|Q)}{\pi_{\theta_{\text{old}}}(o_i|Q)}$ is the likelihood ratio, $A_i$ the advantage, and $\mathrm{D}_{\text{KL}}$ is a KL-divergence penalty to keep the policy close to a reference (supervised fine-tuned) model Ouyang et al. (2022). In practice, $\epsilon$ and $\beta$ are tuned to balance exploration and stability.

**Reward Model Design.** A key differentiator of XYZ-Text2SQL-R1 is its adherence to a minimal, execution-driven reward formulation. While earlier works have often sought highly elaborate reward signals (aggregating string overlap, schema conformance, partial matching, etc. Pourreza et al. (2025); Ma et al. (2025)), we find these can encourage reward hacking and brittle behaviors Skalse et al. (2022).

Table 2: The datasets and their volume used in the paper. For BIRD-training, spider-training/dev, their original sample sizes are 9428, 8659/1034.

| Dataset | Used for Training | | | | Used for Evaluation | | | | | |
|---|---|---|---|---|---|---|---|---|---|---|
| | BIRD training | SPIDER-training | SPIDER-dev | Gretel-Synth Filtered | BIRD-dev | SPIDER-test | Spider2.0-SQLite | Spider-DK | EHR SQL | Science Benchmark |
| Size | 8 017 | 6 972 | 985 | 11 811 | 1534 | 2 147 | 135 | 535 | 1 008 | 299 |

Table 3: Learning from training data (all with GRPO optimization and Qwen2.5-Coder).

| Base Model | Training Data | BIRD-dev | SPIDER-test |
|---|---|---|---|
| 14B-Inst | BIRD, SPIDER | 64.9 | 86.8 |
| 14B-Inst | BIRD, SPIDER, Gretel-Synth-NonFiltered | 64.6 | 86.4 |
| 14B-Inst | BIRD, SPIDER, Gretel-Synth-Filtered | **66.5** | **88.3** |
| 32B-Inst | BIRD, SPIDER, Gretel-Synth-Filtered | 64.9 | 87.7 |
| 32B-Inst | BIRD, SPIDER, Gretel-Synth-Filtered, BIRD-aug-NonFiltered | 62.5 | 86.1 |
| 32B-Inst | BIRD, SPIDER, Gretel-Synth-Filtered, BIRD-aug-Filtered | 64.9 | 86.8 |

Instead, we define a reward function focused solely on (1) *final execution correctness* and (2) *basic syntax validity*:

$$R = \begin{cases} 1, & \text{if the execution results exactly align with ground truth;} \\ 0.1, & \text{if syntax is correct and SQL is executable;} \\ 0, & \text{otherwise.} \end{cases}$$

Here, execution alignment is determined by running the model's SQL prediction alongside the ground-truth query and matching their results using the strictest available criteria (e.g. BIRD benchmark guidelines), ensuring meaningful progress signals. Syntax validity ensures that models are not unduly penalized for benign formatting or minor structural errors when learning to compose well-formed queries. Our design is similar to math/logic RL works Xie et al. (2025); Guo et al. (2025) with extra constraints on valid SQL execution.

The proposed reward design enables stable, interpretable credit assignment, removing distracting or confounding partial rewards. As empirical results (Section 5) demonstrate, our streamlined approach is sufficient and preferable for high-accuracy, generalizable Text2SQL modeling.

## 4 LEARNINGS

Our iterative exploration involved experiments with diverse training and data selection strategies, yielding a variety of empirical insights. While we recognize that some of these observations may be context-specific and not universally applicable, we document them here to inform and accelerate future research in the field.

**Experimental Setup.** Our primary training datasets are derived from the BIRD Li et al. (2024) training set, and the training/development splits of SPIDER Yu et al. (2018). Rather than directly adopting these sources, we apply a filtering process aimed at data quality (see Table 2 and Section 4.1 for details). Initial model selection is guided by performance on the BIRD-dev and SPIDER-test sets, reported via execution accuracy. As we noticed that a sole focus on these two benchmarks risks overfitting, we later broadened our evaluation suite for a more robust assessment.

As base models, we rely on the Qwen2.5-Coder series Hui et al. (2024), including both base, instruct and reasoning-oriented variants. Unless noted otherwise, GRPO Shao et al. (2024) is used as our default RL algorithm. RL-specific settings include a generation temperature of 0.8, a total batch size of 256 (16 rollouts each), an update batch size of 128 per GRPO step, KL penalty $\beta = 0.001$, and clip ratio $\epsilon = 0.2$ (see Section 3 for hyperparameter details).

### 4.1 LEARNINGS FROM TRAINING DATA

**The Critical Role of Filtering.** Thorough inspection of BIRD and SPIDER training splits reveals that many reference SQL queries in these datasets return empty results upon execution. For RL, where

Table 4: Learnings from training strategy (*: limited results because of the checkpoint loss)

| Base Model | Training Strategy | Optimization | BIRD-dev | SPIDER-test |
|---|---|---|---|---|
| Qwen2.5-Coder-32B-Inst | — | GRPO | 64.9 | 87.7 |
| Qwen2.5-Coder-32B-Inst | — | PPO | 63.0 | 85.7 |
| Qwen2.5-Coder-32B | — | GRPO | 64.4 | 87.3 |
| Qwen2.5-Coder-32B-Inst | — | GRPO | 64.9 | 87.7 |
| QwQ-32B | — | GRPO | 55.2 | 79.3 |
| Qwen2.5-Coder-32B-Inst | Batch RL | GRPO | 64.9 | 87.7 |
| Qwen2.5-Coder-32B-Inst * | Online RL | GRPO | 66.6 | – |
| Qwen2.5-Coder-32B-Inst * | Online RL | GRPO | 66.6 | – |
| OmniSQL-32B | Online RL | GRPO | 67.9 | 88.2 |
| OmniSQL-32B | Online RL + Self-defined Prompt Template | GRPO | 67.9 | 88.2 |
| OmniSQL-32B | Online RL + Modified OmniSQL Prompt | GRPO | 70.5 | 88.7 |

reward signaling is tied to execution correctness, such examples can disrupt the learning process by producing spurious or uninformative rewards. We exclude these and filter out samples with execution times exceeding five seconds, markedly reducing overall training time. This straightforward filtering step removed about 1,400 samples from BIRD and 1,700 from SPIDER, yielding a more reliable reward signal and expediting RL convergence. Table 2 provides a summary of datasets used, including derived data splits.

**Table Enhanced Generation with Model-based Data Filtering for Synthetic Data.** We further augment training data with Gretel-Synth Meyer et al. (2024), which provides schemas without populated data. We use GPT-4o to produce INSERT statements per table (details in Appendix B), repeatedly sampling until the reference SQL retrieves non-empty results. Distractor tables from related domains are randomly added to increase schema complexity, and only queries with SQL length > 160 characters and successful execution are retained for the non-filtered pool. However, naively adding Gretel-Synth-NonFiltered to training reduced performance (the first section of Table 3). To remedy this, we employed a model-based filtering, using our best Qwen2.5-Coder-32B-Inst-trained model, and retained only queries where at least one of the ten generations (temperature = 1.0) was correct. This curated Gretel-Synth-Filtered set markedly improves results.

**Unsuccessful Attempts at LLM-Based Data Augmentation.** Drawing inspiration from works such as Yang et al. (2024c); Hu et al. (2023), we attempted to boost diversity through data augmentation, prompting LLMs to paraphrase or generate complex questions given a schema, the original question, and its SQL. Nevertheless, we found that model outputs often closely mirrored the original conditions, limiting diversity. We next tried prompting solely with schema descriptions and gold SQL, omitting the original question, and incorporated self-correction Deng et al. (2025) to ensure all SQLs were executable and returned data. Despite this, augmented data largely failed to improve model generalization (the second section of Table 3), which we attribute to (1) insufficient linguistic and structural diversity, and (2) oversampling schema-specific patterns, leading to overfitting.

**Other Data Sources.** We also experimented with model-filtered data from SynSQL-2.5M Li et al. (2025), but initial attempts were inconclusive. Given the sheer volume of SynSQL-2.5M, we believe more sophisticated filtering could unlock further gains, which we leave for future work.

## 4.2 LEARNINGS FROM TRAINING STRATEGY

Unless stated otherwise, the following rely on filtered BIRD, SPIDER, and Gretel-Synth.

**GRPO vs PPO.** We compare GRPO and PPO. GRPO, by design, reduces memory usage of critic models and is well-suited for large-scale settings; PPO is simpler and historically popular for stable RL optimization. In our experiments, GRPO outperforms PPO by a comfortable margin (see the first section of Table 4), though we caution that PPO may benefit from further hyperparameter tuning.

**Selecting the Optimal Starting Model.** We assess Qwen-2.5-Coder in the base, instructed, and the reasoning-focused QwQ-32B variant. Results show that starting from better instruction following, higher-accuracy models is crucial (see the second section of Table 4). The main differences between Qwen2.5-Coder-32B base and instruction are (1) the instruction following capability and (2) high-quality instruction finetuning data, which leads to about 0.4 EX gap. QwQ-32B is optimized for math and reasoning tasks, e.g., math, but not SQL, and its initial accuracy trailed the

Qwen-2.5-Coder-instructed version by over 10 points, a gap RL fine-tuning could not close. Later, we discuss how strong supervised checkpoints improve downstream RL.

**Online RL Surpasses Batch RL.** We compare online RL, where the model continually interacts with the environment, with batch RL. Our findings show superior results with online RL, likely due to its increased adaptivity and exposure to more complex negative examples via live interaction (the third section of Table 4). This observation extends previous results primarily observed in mathematical and programming, indicating its pertinence for Text2SQL.

**Supervised Fine-Tuning Model Quality Matters.** Stronger SFT models (e.g., OmniSQL Li et al. (2025)) consistently yield better downstream RL results (the forth section of Table 4), reinforcing the importance of strong initializations and echoing observations from other domains.

**Prompt Format is Crucial.** We observed significant gains (the fifth section of Table 4) when switching from a generic prompt to the original OmniSQL prompt, adapted for RL training (see Figure D.1 for prompt). Prompt structure, inclusion of thinking instructions, and database serialization choices all contributed to improved model performance.

**Other Observations.** We explored variations in rollout count (16, 24, 32), human- or LLM-generated prompts, and various reward designs. None produced significant improvement, and, notably, more fine-grained reward designs induced "lazy" behaviors, where models pursued local optima for short-term rewards rather than global correctness.

## 4.3 LEARNING FROM EVALUATION BENCHMARK DIVERSITY

Table 5: Diverse evaluation helps to identify generalization capabilities (OmniSQL-7B).

| Different setting | BIRD-dev | Spider-test | Spider2.0-SQLite | Spider-DK | EHR SQL | Science Benchmark | Average |
|---|---|---|---|---|---|---|---|
| BIRD only | 67.6 | 87.8 | 8.9 | 76.3 | 34.9 | 50.5 | 54.3 |
| BIRD, SPIDER, Gretel-Synth | 67.7 | 88.2 | 11.9 | 79.1 | 35.5 | 51.8 | 55.7 |

**Prompt Optimization Benefits OSS Model Evaluation.** Prompt selection dramatically affects open-source models (e.g., Llama Grattafiori et al. (2024) and Qwen Hui et al. (2024)) performance, with the OmniSQL prompt improving Llama-3.1-70B's BIRD-dev accuracy from 57.4% Zhai et al. (2025) to 65.1% in our experiments. Prompt tuning is thus indispensable for the fair comparison of large, general-purpose LLMs in the Text2SQL task. However, finding the best prompt for general-purpose LLMs is beyond the scope of our work.

**Diverse Evaluation Surfaces Generalization Gaps.** Restricting training to BIRD, we observed strong benchmark accuracy on BIRD-dev, but an average score 1.4 points lower when evaluated across a broader suite (Table 5), including BIRD-dev Li et al. (2024), SPIDER Yu et al. (2018), Spider2.0 Yu et al. (2018), Spider-DK Gan et al. (2021), EHRSQL Lee et al. (2022), and ScienceBenchmark Zhang et al. (2023). This highlights the risk of overfitting and the necessity of evaluating on multiple datasets to ensure robust generalization.

## 5 MAIN RESULT

Our final experiments build upon all prior empirical insights: the training data includes the filtered BIRD-training and SPIDER-training/dev sets, along with the model-based filtered Gretel-Synth-Filtered synthetic examples. The reinforcement learning setup consists of (1) GRPO as the optimization algorithm, (2) online training, (3) initializing from OmniSQL supervised checkpoints, and (4) using a modified OmniSQL prompt for both training and evaluation. We report execution-based accuracy results across all six benchmarks detailed in Table 2.

XYZ-Text2SQL-R1 shows strong performance on the BIRD benchmark (Table 7). Our largest model ranks third overall, but first among models with released research papers. XYZ-Text2SQL-R1-32B reaches 73.84% accuracy with Few-sample Self-Consistency and 71.83% under the stricter greedy decoding setting. In the leaderboard, the Self-Consistency column indicates the number of sampled candidates (e.g., Few: 1–7, Many: 8–32). Both XYZ-Text2SQL-R1-7B and XYZ-Text2SQL-R1-14B

Table 6: Comparison between XYZ-Text2SQL-R1 and other Open-Source Software (OSS) / Proprietary models. *: SQL-R1 Ma et al. (2025) uses majority voting in evaluation. **: (Pourreza et al., 2025, Table 2) might use schema linking, our number follows (Pourreza et al., 2025, Table 3/4). †: Spider2.0 was updated, so we re-evaluate it instead of using the OmniSQL number. For SQL-specific models, generally sensitive to prompt and database serialization, we take the number from their papers. DPSK stands for DeepSeek.

| Model | OSS | BIRD (dev) | SPIDER (test) | Spider2.0-SQLite† | Spider-DK | EHR SQL | Science Benchmark | Average |
|---|---|---|---|---|---|---|---|---|
| **Models Size < 10B** | | | | | | | | |
| DPSK-Coder-6.7B-Instruct Guo et al. (2024) | ✓ | 43.1 | 70.5 | 4.4 | 60.9 | 28.6 | 40.8 | 41.4 |
| Qwen2.5-Coder-7B-Instruct Hui et al. (2024) | ✓ | 50.9 | 82.2 | 4.4 | 67.5 | 24.3 | 45.2 | 45.8 |
| Qwen2.5-7B-Instruct Yang et al. (2024a) | ✓ | 46.9 | 76.8 | 5.2 | 63.7 | 20.9 | 38.5 | 42.0 |
| SQL-R1-7B* Ma et al. (2025) | ✗ | 66.6 | – | – | – | – | – | – |
| OmniSQL-7B Li et al. (2025) | ✓ | 63.9 | 87.9 | 13.3 | 76.1 | 34.9 | 50.2 | 54.4 |
| Think2SQL-7B Papicchio et al. (2025) | ✗ | 56.1 | – | – | – | – | – | – |
| OpenCoder-8B-Instruct Huang et al. (2024) | ✓ | 37.5 | 68.3 | 1.5 | 62.6 | 21.9 | 39.8 | 38.6 |
| Meta-Llama-3.1-8B-Instruct Grattafiori et al. (2024) | ✓ | 42.0 | 72.2 | 1.5 | 62.6 | 24.6 | 36.8 | 40.0 |
| Granite-8B-Code-Instruct Mishra et al. (2024) | ✓ | 27.6 | 64.9 | 1.5 | 50.7 | 16.0 | 29.4 | 31.7 |
| Granite-3.1-8B-Instruct Mishra et al. (2024) | ✓ | 36.0 | 69.8 | 3.7 | 60.0 | 19.6 | 36.8 | 37.7 |
| **XYZ-Text2SQL-R1-7B** | ✓ | **68.9** | **88.8** | **15.6** | **81.5** | **36.7** | **51.8** | **57.2** |
| **10B ≤ Models Size ≤ 30B** | | | | | | | | |
| Qwen2.5-Coder-14B-Instruct Hui et al. (2024) | ✓ | 61.5 | 86.6 | 11.1 | 73.6 | 31.6 | 52.2 | 52.8 |
| Qwen2.5-14B-Instruct Yang et al. (2024a) | ✓ | 56.7 | 82.0 | 8.1 | 72.3 | 28.8 | 51.2 | 49.9 |
| OmniSQL-14B Li et al. (2025) | ✓ | 64.2 | 88.3 | 12.6 | 72.9 | 39.9 | 56.9 | 55.8 |
| Reasoning-SQL-14B** Pourreza et al. (2025) | ✗ | 64.2 | 81.4 | – | 73.0 | – | – | – |
| Starcoder2-15B-Instruct Lozhkov et al. (2024) | ✓ | 38.5 | 73.0 | 1.5 | 66.5 | 16.8 | 25.8 | 37.0 |
| DPSK-Coder-V2-Inst (16B/MoE) Zhu et al. (2024a) | ✓ | 44.6 | 77.9 | 2.2 | 63.7 | 23.9 | 39.1 | 41.9 |
| Granite-20B-Code-Instruct Mishra et al. (2024) | ✓ | 34.0 | 74.1 | 1.5 | 62.2 | 23.5 | 37.5 | 38.8 |
| Codestral-22B Mistral AI (2024) | ✓ | 52.7 | 78.6 | 8.1 | 69.9 | 37.8 | 48.5 | 49.3 |
| **XYZ-Text2SQL-R1-14B** | ✓ | **70.1** | **89.4** | **16.3** | **79.4** | **40.7** | **58.2** | **59.0** |
| **30B < Models Size or Unknown** | | | | | | | | |
| Qwen2.5-Coder-32B-Instruct Hui et al. (2024) | ✓ | 64.5 | 87.5 | 10.4 | 78.3 | 36.4 | 54.8 | 55.3 |
| Qwen2.5-32B-Instruct Yang et al. (2024a) | ✓ | 62.0 | 84.9 | 10.4 | 73.1 | 33.6 | 50.5 | 52.4 |
| Xiyan-SQL-32B Gao et al. (2024) | ✓ | 67.0 | – | – | – | – | – | – |
| ExDPO-32B Zhai et al. (2025) | ✓ | 68.3 | 85.1 | – | – | – | – | – |
| OmniSQL-32B Li et al. (2025) | ✓ | 64.5 | 87.6 | 14.8 | 76.1 | 42.4 | 57.2 | 57.1 |
| DPSK-Coder-33B-Instruct Guo et al. (2024) | ✓ | 49.2 | 74.3 | 5.2 | 69.0 | 31.4 | 44.5 | 45.6 |
| Granite-34B-Code-Instruct Mishra et al. (2024) | ✓ | 33.8 | 74.4 | 0.0 | 64.7 | 23.8 | 40.1 | 39.5 |
| Mixtral-8x7B-Inst. (47B, MoE) Jiang et al. (2024) | ✓ | 35.3 | 67.8 | 3.7 | 55.3 | 21.5 | 29.4 | 35.5 |
| Meta-Llama-3.1-70B-Instruct Grattafiori et al. (2024) | ✓ | 65.1 | 84.3 | 7.4 | 75.1 | 37.4 | 55.2 | 54.1 |
| ExDPO-70B Zhai et al. (2025) | ✓ | 68.5 | 86.6 | – | – | – | – | – |
| Qwen2.5-72B-Instruct Yang et al. (2024a) | ✓ | 60.3 | 84.0 | 11.9 | 76.4 | 35.0 | 52.8 | 53.4 |
| Command-A-111B Team Cohere (2025) | ✓ | 63.5 | – | – | – | – | – | – |
| DeepSeek-V3 (671B, MoE) Liu et al. (2024) | ✓ | 63.2 | 85.5 | 14.8 | 72.9 | 43.2 | 56.2 | 55.6 |
| GPT-4o-mini OpenAI (2023) | ✗ | 58.8 | 82.4 | 11.9 | 73.3 | 37.9 | 51.8 | 52.7 |
| GPT-4-Turbo OpenAI (2023) | ✗ | 62.0 | 83.4 | 13.3 | 72.3 | 43.1 | 59.2 | 55.6 |
| GPT-4o OpenAI (2023) | ✗ | 61.9 | 83.2 | **17.0** | 72.9 | **44.9** | 55.5 | 55.9 |
| **XYZ-Text2SQL-R1-32B** | ✓ | **70.5** | **88.7** | 16.3 | **80.6** | 40.1 | **60.9** | **59.5** |

Table 7: BIRD Single-Model Leaderboard.

| Model | Self-Consistency | Dev | Test |
|---|---|---|---|
| Databricks RLVR 32B | Few | – | 75.7 |
| Sophon-Text2SQL-32B | Many | 72.4 | 74.8 |
| XYZ-Text2SQL-R1-32B | Few | 72.2 | 73.8 |
| Databricks RLVR 32B | – | 70.8 | 73.6 |
| Jiayin-Pangu-Text2SQL-14B | Many | 71.1 | 73.5 |
| XYZ-Text2SQL-R1-14B | Few | 71.4 | 72.2 |
| SIFT-32B | Scale | 70.1 | 70.9 |
| CrazyData-Text2SQL-32B | Many | – | 70.7 |
| Infly-RL-SQL-32B | Few | 70.1 | 70.6 |
| XYZ-Text2SQL-R1-7B | Few | 70.1 | 70.4 |

Table 8: XYZ-Text2SQL-R1 with different evaluation techniques on BIRD-dev.

| Model Size | Value Retrieval | Majority Voting | BIRD-dev |
|---|---|---|---|
| 14B | ✗ | ✗ | 70.1 |
| | ✓ | ✗ | 70.5 |
| | ✗ | ✓ | 70.6 |
| | ✓ | ✓ | 70.8 |
| 32B | ✗ | ✗ | 70.5 |
| | ✓ | ✗ | 70.9 |
| | ✗ | ✓ | 71.2 |
| | ✓ | ✓ | 71.5 |

exceed 70% on the BIRD test, and notably, XYZ-Text2SQL-R1-7B matches the performance of much larger systems such as ExCoT-70B Zhai et al. (2025) with an order of magnitude fewer parameters.

## 5.1 PERFORMANCE EVALUATION ACROSS SIX BENCHMARKS

Instead of optimizing the model for single benchmark performance, XYZ-Text2SQL-R1 achieves best-in-class average performance across six different benchmarks. Table 6 presents a comprehensive comparison between XYZ-Text2SQL-R1 and a wide range of open-source and proprietary models,

including general-purpose LLMs (e.g. DeepSeek-V3, GPT-4o, GPT-4-Turbo), coding models (Qwen-Coder), and specialized Text2SQL models (XiYan-SQL, OmniSQL, ExCoT, SQL-R1, Reasoning-SQL). Except where otherwise indicated, all evaluations use single-model, single-inference (i.e., greedy decoding with no ensembling, schema linking, or external re-ranking).

**State-of-the-Art Across All Scales.** XYZ-Text2SQL-R1 consistently outperforms existing state-of-the-art models across all parameter scales and benchmarks. For models under 10B parameters, XYZ-Text2SQL-R1-7B achieves an average accuracy of 57.2, outperforming open-source competitors (e.g., OmniSQL-7B at 54.4) and closely rivaling, or surpassing, closed-source models such as SQL-R1-7B. In the 10B–30B range, XYZ-Text2SQL-R1-14B attains 59.0, exceeding strong baselines like OmniSQL-14B and Reasoning-SQL-14B. In the 30B+ category, XYZ-Text2SQL-R1-32B achieves the best results overall, with an average of 59.5, outperforming the largest open-source and commercial models, including DeepSeek-V3 (55.6), GPT-4-Turbo (55.6), and even GPT-4o (55.9).

**Parameter Efficiency and Task Specialization.** Notably, XYZ-Text2SQL-R1-7B matches or surpasses models such as DeepSeek-V3 (671B, MoE) and OmniSQL-32B on six benchmark accuracy, despite a fraction of their parameter count. These results highlight the advantages of task-specific training and reward design for compositional SQL reasoning.

**Benchmark-Specific Highlights.** On challenging benchmarks such as BIRD, Spider, and EHRSQL, XYZ-Text2SQL-R1 shows marked improvements. For instance, on the BIRD-dev split, XYZ-Text2SQL-R1 7B, 14B, and 32B reach 68.9, 70.1, and 70.5, setting new records across size categories. Substantial gains are also seen on Spider-DK and the Science benchmark, underscoring XYZ-Text2SQL-R1's robustness and domain coverage.

Table 9: Generalization of XYZ-Text2SQL-R1 across different model families.

| Model | BIRD-dev | Spider-test | Spider2.0-SQLite | Spider-DK | EHR SQL | Science Benchmark | Average |
|---|---|---|---|---|---|---|---|
| Qwen3-1.7B-Instruct | 46.6 | 77.4 | 2.2 | 63.2 | 19.9 | 42.1 | 41.9 |
| + XYZ-Text2SQL-R1 | $58.7_{+12.1}$ | $83.8_{+6.4}$ | $4.4_{+2.2}$ | $72.3_{+9.1}$ | $33.9_{+14.0}$ | $47.8_{+5.7}$ | $50.2_{+8.3}$ |
| LLaMA-3.2-3B-Instruct | 18.2 | 50.3 | 0.7 | 34.4 | 5.5 | 20.7 | 21.6 |
| + XYZ-Text2SQL-R1 | $46.9_{+28.7}$ | $79.6_{+29.3}$ | $5.2_{+4.5}$ | $66.2_{+31.8}$ | $26.2_{+20.7}$ | $40.1_{+19.4}$ | $44.0_{+22.4}$ |

**Generalizability Across Model Families.** To demonstrate that the effectiveness of XYZ-Text2SQL-R1 is not confined to the Qwen-Coder family, we applied our training framework to two additional open-source models: Qwen3-1.7B-Instruct Yang et al. (2025a) and LLaMA-3.2-3B-Instruct Dubey et al. (2024). As shown in Table 9, we observed an average absolute improvement of 8.3% for Qwen3-1.7B and a remarkable 22.4% for LLaMA-3.2-3B. We also noted that our framework helps align models with specific output formatting instructions, which contributed to the significant gains for LLaMA-3.2-3B. These results underscore the robustness and generalizability of our training paradigm across diverse model families.

## 6 DISCUSSION

Beyond the benchmark achievements, the development of XYZ-Text2SQL-R1 revealed some insights into model behavior and potential for exploiting inference-time enhancements. These also motivate further examination of the model's comparative standing.

**Enhancing XYZ-Text2SQL-R1 with Other Techniques.** While XYZ-Text2SQL-R1 is designed for efficient, direct inference, it can also serve as the SQL generator in more complex agentic systems. Table 8 shows results using two inference-time enhancements: value retrieval (from Talaei et al. (2024)) and majority voting (as in Li et al. (2025); Ma et al. (2025)) with eight generations per sample. Each technique yields a tangible improvement; combined, they boost XYZ-Text2SQL-R1-32B performance by up to

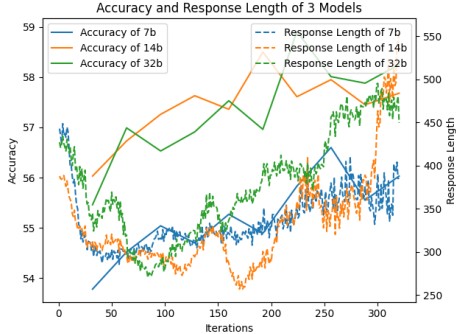

Figure 1: Generation length and average accuracy across six benchmarks.

one point on BIRD-dev, demonstrating complementary gains from orthogonal reasoning components.

Table 10: The comparison between XYZ-Text2SQL-R1 and Reasoning-SQL Pourreza et al. (2025). The numbers for the latter from (Pourreza et al., 2025, Tables 4 and 5) and it might involve schema linking as mentioned in (Pourreza et al., 2025, Table 2), but we are unsure; see Table 1 for the reward design comparison.

| Method | Base Model | Filtering Method | Reward Design | BIRD -dev | SPIDER -test | SPIDER -DK |
|---|---|---|---|---|---|---|
| Reasoning-SQL | Qwen2.5-Coder-14B-Inst | Large Models Prompt | Complex | 64.21 | 81.43 | 73.03 |
| XYZ-Text2SQL-R1 | Qwen2.5-Coder-14B-Inst | Empty Golden-SQL | Simple | 66.49 | 87.20 | 75.10 |

Table 11: Comparison of XYZ-Text2SQL-R1, SQL-R1 Ma et al. (2025) and Think2SQL Papicchio et al. (2025). GRE: greedy decoding, MV: Majority Voting (8 candidates). Competitors numbers according to (Ma et al., 2025, Table 2) and (Papicchio et al., 2025, Table 1 and 3); see Table 1 for the reward design comparison.

| Method | Base Model | SFT (#Samples) | Reward Design | BIRD-dev GRE/MV | SPIDER-test GRE/MV |
|---|---|---|---|---|---|
| SQL-R1 | Qwen2.5-Coder-7B-Inst | ✗ | Complex | –/63.1 | –/86.1 |
| SQL-R1 | Qwen2.5-Coder-7B-Inst | ✓(200K) | Complex | –/59.2 | –/86.4 |
| Think2SQL | Qwen2.5-Coder-7B-Inst | ✓(9K) | Complex | 56.1/– | 82.4/– |
| XYZ-Text2SQL-R1 | Qwen2.5-Coder-7B-Inst | ✗ | Simple | **63.9/64.8** | **85.0/87.1** |
| SQL-R1 | OmniSQL-7B | ✗ | Complex | –/66.6 | –/**88.7** |
| XYZ-Text2SQL-R1 | OmniSQL-7B | ✗ | Simple | **67.6/69.4** | **87.8**/88.6 |

**Response Length vs. Accuracy.** Figure 1 plots the evolution of average response length and execution accuracy across training for all three sizes of XYZ-Text2SQL-R1. We observe a U-shaped pattern in response length—initially decreasing, then gradually increasing—as accuracy rises. This trend may reflect early-stage conservative decoding (and possible undertraining), followed by richer, more grounded outputs as training progresses, an effect also noted in some of the previous works Xie et al. (2025).

**Why RL is Effective in Text2SQL.** Unlike supervised fine-tuning, which relies solely on positive examples, RL enables the model to learn from failures through trial and error with negative feedback. This allows the model to overcome inherent ambiguities in the training data and better capture user intent. Appendix F provides qualitative examples.

**Reasoning-SQL.** To enable a fair comparison with Reasoning-SQL Pourreza et al. (2025), we retrain XYZ-Text2SQL-R1 from the Qwen-2.5-14B-Instruct base, use BIRD-training data alone, and run 3 epochs. Under these controlled settings, the two methods differ primarily in training data filtering (XYZ-Text2SQL-R1 uses simple empty-return-based removal, Reasoning-SQL uses LLM filtering) and reward design (our simple execution/syntax signal vs. a complex mixture). As shown in Table 10, XYZ-Text2SQL-R1 outperforms Reasoning-SQL by clear margins on BIRD-dev (+2.28), Spider-test (+5.77), and Spider-DK (+2.07). This underscores the power of direct, execution-guided reinforcement learning and streamlined filtering.

**SQL-R1 and Think2SQL.** We further compare XYZ-Text2SQL-R1 with recently published SQL-R1 Ma et al. (2025) and Think2SQL Papicchio et al. (2025), restricting RL fine-tuning to BIRD-training and evaluating under both greedy and majority voting regimes. As summarized in Table 11, XYZ-Text2SQL-R1 delivers the best or near-best performance in all configurations and consistently outperforms prior approaches, even without extensive pre-training or complex reward engineering.

## 7 CONCLUSIONS

We presented XYZ-Text2SQL-R1, a novel RL framework for Text2SQL that uses GRPO and a simple execution-based reward. It achieves 71.83% execution accuracy (73.84% with self-consistency) on BIRD-test as a single 32B model, ranking among the top leaderboard entries while surpassing substantially larger LLMs. Across six challenging benchmarks, XYZ-Text2SQL-R1 yields up to 4 points improvement over strong baselines, with notable parameter efficiency (e.g., the 7B version matches or outperforms prior ExCoT-70B models and GPT4o). Our results highlight that minimal reward signals, principled data filtering, and carefully curated training strategies are key for robust SQL generation. We release models to support future research in this area.

## ETHICS STATEMENT

This work complies with the ICLR Code of Ethics. We use only publicly available datasets (BIRD, Spider, EHRSQL, ScienceBenchmark), with no collection of private or personally identifiable information. No human subjects or sensitive data were involved. Potential ethical concerns relate to biases in benchmark datasets. This research is for academic purposes only, and all experiments were conducted in accordance with standard practices of research integrity.

## REPRODUCIBILITY STATEMENT

We provide detailed descriptions of datasets, filtering procedures, model training, and evaluation benchmarks in the main text (Sections 3, 4, 5). Appendices include prompt templates, data augmentation pipelines, and qualitative case studies. LLMs were used only for language polishing, not for research design or data generation. The code, datasets, and models will be released after paper acceptance under the supervision of institutional policies, to support independent verification and future research.

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

## A    LLM Usage Statement

In preparing this manuscript, we employed LLMs solely for language polishing. Specifically, we used commercial LLMs as writing assistants to improve sentence structure, check grammar and spelling, and enhance overall readability and clarity of the text. No LLMs were used for research ideation, methodological design, data analysis, or code generation. All scientific contributions, including theoretical development, experimental design, implementation, and analysis, were solely conducted by the authors.

All text outputs from LLMs were carefully reviewed and revised by the authors, who take full responsibility for the accuracy, originality, and integrity of the manuscript. LLMs are not considered authors and bear no responsibility for this work.

## B    Construction Pipeline from Gretel-Synth to Gretel-Synth-NonFiltered

To enable SQL execution and ensure meaningful query results, we use a carefully designed prompt to guide an LLM in generating realistic table data. Appendix B.1 illustrates this prompt, which produces valid `INSERT` statements aligned with both the SQL context and query.

These synthetic data inserts serve as the foundation for the construction pipeline outlined in Algorithm 1, which augments each sample with distractive tables and applies filtering to retain executable, informative SQL examples.

## C    Data Augmentation

We perform data augmentation on the BIRD dataset following the prompt template shown in Figure C.1. Note that the *task* component is optional. To encourage diversity in the generated data, we omit the *task* and *answer* fields when necessary.

After obtaining a set of SQL queries, we apply a self-correction workflow as described in Algorithm 2 to ensure the SQLs are executable and yield non-empty results. The prompts used for `self_correct` and `similar_error_refine` are provided in Figure C.2.

## D    Prompt Used For Training/Evaluation

We present the prompt used for our training and evaluation in Figure D.1.

## E    Examples from XYZ-Text2SQL-R1-32B

We provide several examples here to illustrate the thinking process of XYZ-Text2SQL-R1-32B in Figure E.1, E.2, and E.3. Note that we did not cherry-pick the answer so the generated SQL might be wrong.

## F    Qualitative Case Studies of RL Training Effects

To better illustrate how RL with simple reward signals improves the reasoning capabilities of our model, we present two detailed case studies. These examples highlight the unique advantages of our framework in correcting subtle but critical reasoning errors that are difficult to address through supervised fine-tuning alone.

**Case Study 1: Resolving Ambiguity Between Aggregation and Enumeration.** Consider the following question: *"How many female patients born in 1964 were admitted to the hospital? List them by ID."* This query is inherently ambiguous because it mixes two operations: counting patients (COUNT(ID)) and enumerating their identifiers (SELECT ID). A model without RL training, such as OmniSQL-32B Li et al. (2025), often attempts to reconcile these conflicting intents by generating a

---

**Prompt for Adding Synthetic Data**

**Adding Synthetic Data Prompt**
You are an expert in SQL data modeling. Your task is to analyze the given SQL schema and, if necessary, generate realistic and logically consistent sample data to ensure:
For a given <SQL Prompt>, both <SQL Query> and <SQL Context> can meet its requirements, and <SQL Query> can query the corresponding data from the TABLE created by <SQL Context>.

Given a – **<SQL Prompt>**:
{question}

I have generated the <SQL Query> and <SQL Context>:
– **<SQL Query>**:
{sql_query}

– **<SQL Context>**:
{sql_context}

{error_infomations}

I need data samples to validate the correctness of the <SQL Query>.
Therefore, please help me add one `INSERT` statement for each table in the <SQL Context>, with 5 sample rows per table.
The inserted data should ensure that the <SQL Query> can retrieve results from the tables.
Please ensure that it does not cause errors when using `sqlite3`.
Please do not include any additional explanations or instructions.

Please help me fix this **<SQL Context>** and ensure that it contains at most five records.
Please also help me modify **<SQL Query>** to ensure that it does not cause errors when using `sqlite3`.

Please give your expanded **<SQL Context>** in: \\sql_context
your fixed **<SQL Query>** in: \\sql_query
and the **INSERT statements** in: \\sql_insert

---

Figure B.1: Prompt for Generating Executable SQL Context and Synthetic Data Inserts in the Gretel-Synth Pipeline

`UNION` query that merges an aggregate value (a single count) with a multi-row column (a list of IDs), as shown in Example F.1. Such an output is either syntactically invalid or semantically incoherent, rendering it useless. In contrast, our RL-trained model learns to avoid these pitfalls and instead produces a correct and executable query, as demonstrated in Example F.2. During training, erroneous attempts of this kind consistently fail to execute and thus receive negative rewards. Over time, this feedback teaches the model both the grammatical constraints of SQL and, more importantly, the need to prioritize the user's actual intent (in this case, "List them").

**Case Study 2: Filtering Out Contextual Noise.** A second source of errors arises from semantic misinterpretation. In the BIRD Li et al. (2024) dataset, natural language questions sometimes include contextual metadata such as the database name (e.g., *student_club*). Non-RL models frequently misinterpret this metadata as a core entity, leading to reasoning errors such as searching for a literal club named "student_club," as illustrated in Example F.3. Our RL framework penalizes such misinterpretations whenever they produce incorrect execution results. Through repeated negative feedback, the model learns to differentiate between background context and the true semantic target of the query. Consequently, the RL-trained model is able to maintain focus on the actual user request, as shown in Example F.4, rather than being misled by irrelevant contextual tokens.

---

**Algorithm 1:** Construction of `Gretel-Synth-NonFiltered`: Inserting Synthetic Data, Adding Distractive Tables, and Filtering Executable SQL Samples

---

**Input:** Full dataset `Gretel-Synth`
**Output:** `Gretel-Synth-NonFiltered` after adding synthetic data and distractive tables
**1. Insert Data Using LLM**
Initialize `Gretel-Synth-NonFiltered` as empty list;
**foreach** sample *in* `Gretel-Synth` **do**
    Initialize `Gretel-Synth-ForNextRound` ← [sample];
    **for** $i \leftarrow 1$ **to** $8$ **do**
        Initialize `next_round` as empty list;
        **foreach** sample *in* `Gretel-Synth-ForNextRound` **do**
            `/* We use GPT-4o to generate insert statements; prompt is provided in Appendix B.1 */`
            `insert_stmts` ← `GPT_generate_inserts(sample)`;
            Execute `sql_context` to create tables;
            Execute `insert_stmts` to populate data;
            `results` ← `execute_sql_query(sample.sql_query)`;
            **if** `results` *not empty* **then**
                Append sample to `Gretel-Synth-NonFiltered`;
            **else**
                Append sample to `next_round`;
        `Gretel-Synth-ForNextRound` ← `next_round`;

**2. Add Distractive Tables**
**foreach** sample *in* `Gretel-Synth-NonFiltered` **do**
    `schema` ← `sample.sql_context`;
    `domain` ← `sample.domain`;
    `existing_tables` ← `extract_table_names(schema)`;
    **if** `existing_tables` *is None* **then**
        **continue**;
    `/* Sample table count from BIRD/Spider distribution with added uniform noise */`
    `n` ← `sample_with_noise_from_table_distribution(table_counts)`;
    `/* Select non-conflicting table schemas from the same domain */`
    `distractive_schemas` ← `select_nonconflicting_schemas(domain, existing_tables, n - 1)`;
    `/* Create distractive tables first, then target tables which may overwrite on name conflict */`
    `create_database(distractive_schemas + schema)`;

**3. Final Selection**
**foreach** sample *in* `Gretel-Synth-NonFiltered` **do**
    **if** *length of* `sample.sql_query` > *160* **and** `sample.sql_query` *returns results* **then**
        Keep sample in `Gretel-Synth-NonFiltered`;
    **else**
        Discard sample from `Gretel-Synth-NonFiltered`;
**return** `Gretel-Synth-NonFiltered`;

---

**Broader Implications.** These case studies illustrate that while non-RL models can perform chain-of-thought reasoning, their reasoning traces are often fragile and prone to subtle errors. SFT exposes the model only to correct examples, showing it "what to do." RL, by contrast, exposes the model to both success and failure, teaching it "what works" and "what does not." The simple binary signals from our GRPO training, reward for success, penalty for failure, are sufficient to correct a wide spectrum of reasoning errors, ranging from technical syntax issues to deeper semantic misinterpretations.

## G    COMPUTE RESOURCE

All experiments were conducted on 8 H100-80GB nodes (a total of 64 H100-80GB GPUs). Training time ranged from a few hours for the 7B model to up to 2.5 days for the final 32B model experiments.

---

**Prompt for BIRD Data Augmentation**

Table information:
Table name: {table name}
Column name: {column name}
Column description: {column desc}
Sample rows: {samples}

Optional(Task: {task}. The answer to the task is: {answer}.)

Based on this, write 10 more complex nested SQLite SQL queries or SQLs with CTEs in sql code block format. You can use any information in the database information provided. Each query should be different. You can write SELECT query only. For each query, just write one sentence to describe the task. Format like:

/*Task: {task description in one sentence}*/
SELECT ...

Don't output other contents.

---

Figure C.1: Prompt for BIRD Data Augmentation

---

**Algorithm 2:** Self-Correction Workflow: Execution of multiple SQLs with self-correction and refinement based on invalid result feedback to ensure augmented SQLs with valid results.

---

**Input:** List of SQL statements `sqls`
**Output:** List of successful results `result_dic_list`
Initialize `result_dic_list` as empty list;
Initialize `error_rec` as empty list;
**while** `sqls` *not empty* **do**
    `sql = sqls[0]`;
    `results = execute_sql_sqlite(sql)`;
    **if** `results` *is valid (i.e., string and not empty)* **then**
        Append `sql` and `results` to `result_dic_list`;
        Continue to next SQL;
    Initialize `max_try`;
    **while** `results` *is not valid* **do**
        **if** `max_try == 0` **then**
            **break**;
        `corrected_sql ← self_correct(sql, results)`;
        **if** `corrected_sql` *is not valid* **then**
            **continue**;
        Execute corrected SQL; `results = execute_sql_sqlite(sql)`;
        Decrease `max_try` by 1;
    **if** `results` *is valid* **then**
        **if** `sqls` *not empty* **then**
            `sqls ← similar_error_refine(sqls)`;
        **if** `corrected_sql` *exists* **then**
            Append `corrected_sql` and `results` to `result_dic_list`;
**return** `result_dic_list`;

---

## H    LIMITATIONS

It is important for each study—especially in the LLM domain—to clearly state its limitations. In our work, as noted in Section 4.2, we did not exhaustively explore PPO hyperparameters, which may limit the strength of our conclusions. Similarly, we only partially examined data augmentation

---

**Prompt for Self-Correction Workflow**

**Self-Correction Prompt**
Input SQL: {sql}
The error information is: {error}
Please correct the SQL based on the previous context. Output your reasoning process followed by only one corrected SQL query in the following format:
– Description: ...
<Corrected SQL here>
Do not output multiple SQLs or only an analysis without a final SQL.

**Similar Error Refinement Prompt**
The following SQL has been corrected:
Original SQL: {sql}
Corrected SQL: {corrected_sql}
Please correct the remaining SQL statements if they contain similar errors. The list of SQLs to be refined is: {sqls}
For each corrected SQL, respond in the following format:
– Description: ...
<Corrected SQL here>

---

Figure C.2: Prompt for Self-Correction Workflow

strategies Section 4.1. In addition, while general-purpose LLMs are known to be prompt-sensitive, we did not systematically explore prompt variations Section 4.3. Finally, we have conducted only limited evaluations of XYZ-Text2SQL-R1 across different model families, without covering broader types and scales of models.

---

**Prompt Template for Training/Evaluation**

**System:**
You are a data science expert. Below, you are provided with a database schema and a natural language question. Your task is to understand the schema and generate a valid SQL query to answer the question.

**User:**
Database Engine:
SQLite

Database Schema: {Database Schema}
This schema describes the database's structure, including tables, columns, primary keys, foreign keys, and any relevant relationships or constraints.

Question:
{evidence + question}

Instructions:
- Make sure you only output the information that is asked in the question. If the question asks for a specific column, make sure to only include that column in the SELECT clause, nothing more.
- The generated query should return all of the information asked in the question without any missing or extra information.
- Before generating the final SQL query, please think through the steps of how to write the query.

Output Format:
Please provide a detailed chain-of-thought reasoning process and include your thought process within '<think>' tags. Your final answer should be enclosed within '<answer>' tags.

Ensure that your SQL query follows the correct syntax and is formatted as follows:

```sql
– Your SQL query here
```

Example format:
<think> Step-by-step reasoning, including self-reflection and corrections if necessary. [Limited by 4K tokens] </think>
<answer> Summary of the thought process leading to the final SQL query. [Limited by 1K tokens]

```sql
Correct SQL query here
```
</answer>

**Assistant:**
Let me solve this step by step.
<think>

---

Figure D.1: Prompt Template for Training and Evaluation

1080
1081
1082
1083
1084
1085
1086
1087
1088
1089
1090
1091
1092
1093
1094
1095
1096
1097
1098
1099
1100
1101
1102
1103
1104
1105
1106
1107
1108
1109
1110
1111
1112
1113
1114
1115
1116
1117
1118
1119
1120
1121
1122
1123
1124
1125
1126
1127
1128
1129
1130
1131
1132
1133

---

**Example I of response from XYZ-Text2SQL-R1-32B**

**Question:** Eligible free rate for K-12 = 'Free Meal Count (K-12)' / 'Enrollment (K-12)'
What is the highest eligible free rate for K-12 students in the schools in Alameda County?

**Assistant:**
Let me solve this step by step.
<think>
1. **Identify the goal**: We need to find the highest eligible free rate for K-12 students in the schools located in Alameda County. The rate is calculated as 'Free Meal Count (K-12)' divided by 'Enrollment (K-12)'.

2. **Determine the relevant table and columns**:
- The relevant table is 'frpm' because it contains the columns 'Free Meal Count (K-12)' and 'Enrollment (K-12)'.
- However, we need to filter by County, which is in the 'schools' table. Thus, we will need to join the 'frpm' table with the 'schools' table on the 'CDSCode' column to filter by the county name.

3. **Formulate the query**:
- We will calculate the eligible free rate by taking the ratio of 'Free Meal Count (K-12)' and 'Enrollment (K-12)'.
- The 'schools' table will be used to filter for schools in Alameda County.
- The 'MAX' function will be used to find the highest eligible free rate.

4. **Join tables**:
- Use an 'INNER JOIN' between the 'schools' table and the 'frpm' table on the 'CDSCode' column to link them correctly.
</think>

<answer>
Here is the correct SQL query:

```sql
SELECT MAX(f.'Free Meal Count (K-12)' / f.'Enrollment (K-12)') AS HighestEligible-
FreeRate
FROM frpm AS f
JOIN schools AS s ON f.CDSCode = s.CDSCode
WHERE s.County = 'Alameda';
```

*Explanation**:
- The 'JOIN' is used to connect the 'frpm' and 'schools' tables on the 'CDSCode' column.
- The 'WHERE' clause filters the records to only include those from Alameda County.
- The 'MAX' function is used to find the highest value of the eligible free rate, which is calculated as 'Free Meal Count (K-12)' divided by 'Enrollment (K-12)'.

The resulting query is efficient and meets the requirements of the natural language question.
</answer>

Figure E.1: Example I of response from XYZ-Text2SQL-R1-32B

**Example II of response from XYZ-Text2SQL-R1-32B**

**Question:** Valid charter number means the number is not null. Rank schools by their average score in Writing where the score is greater than 499, showing their charter numbers.

**Assistant:**
Let me solve this step by step.
<think>
1. **Understanding the Question**:
- We need to rank schools based on their average score in writing ('AvgScrWrite') for scores greater than 499.
- We should display the 'CharterNum' of the school.
- A valid charter number means the number is not null.

2. **Identifying the Necessary Tables**:
- The 'satscores' table contains the average scores for writing ('AvgScrWrite') and the 'cds' code which can be used to join with the 'schools' table.
- The 'schools' table contains the 'CharterNum' field which holds the charter number of the schools.
- We need a join between the 'satscores' and 'schools' tables on the 'cds' (from 'satscores') and 'CDSCode' (from 'schools') fields.

3. **Filter Conditions**:
- We need to filter out schools with 'AvgScrWrite' greater than 499.
- The 'CharterNum' should not be null.

4. **Sorting**:
- The result needs to be sorted by the average score in writing ('AvgScrWrite'), in descending order.

5. **Selecting Columns**:
- We are only interested in the 'CharterNum' field from the 'schools' table.

</think>

<answer>

```sql
SELECT T2.CharterNum
FROM satscores AS T1
INNER JOIN schools AS T2
ON T1.cds = T2.CDSCode
WHERE T1.AvgScrWrite > 499 AND T2.CharterNum IS NOT NULL
ORDER BY T1.AvgScrWrite DESC;
```

</answer>

This query will:
- Join the 'satscores' and 'schools' tables based on the 'cds' and 'CDSCode' fields.
- Filter out schools with an average writing score ('AvgScrWrite') greater than 499 and where 'CharterNum' is not null.
- Sort the results by 'AvgScrWrite' in descending order.
- Return only the 'CharterNum' field.

Figure E.2: Example II of response from XYZ-Text2SQL-R1-32B

---

**Example III of response from XYZ-Text2SQL-R1-32B**

**Question:** Eligible free rates for students aged 5-17 = 'Free Meal Count (Ages 5-17)' / 'Enrollment (Ages 5-17)'
What is the free rate for students between the ages of 5 and 17 at the school run by Kacey Gibson?

**Assistant:**
Let me solve this step by step.
<think>
1. The question asks for the rate of students between the ages of 5 and 17 who are eligible for free meals. The formula provided for this rate is 'Free Meal Count (Ages 5-17) / Enrollment (Ages 5-17)'.
2. From the database schema provided, the relevant data can be found in the 'frpm' table, which contains the columns 'Free Meal Count (Ages 5-17)' and 'Enrollment (Ages 5-17)'.
3. The condition for the school is based on an administrator's name, Kacey Gibson. This information is stored in the 'schools' table in the columns 'AdmFName1' and 'AdmLName1'.
4. To link the 'frpm' table to the 'schools' table, we use the 'CDSCode' column, which is the primary key in both tables.
5. The final query needs to join the 'frpm' and 'schools' tables on their common 'CDSCode' key, filter the rows for the specific administrator, and then compute and return the eligible free rate for students aged 5-17.
</think>
<answer>
```sql
SELECT
(T1.'Free Meal Count (Ages 5-17)' / T1.'Enrollment (Ages 5-17)') AS free_rate
FROM
frpm AS T1
JOIN
schools AS T2
ON
T1.CDSCode = T2.CDSCode
WHERE
T2.AdmFName1 = 'Kacey' AND T2.AdmLName1 = 'Gibson';
```

This SQL query is correct and directly answers the question with the necessary steps and logic based on the provided schema.
</answer>

Figure E.3: Example III of response from XYZ-Text2SQL-R1-32B

---

**Example IV (Incorrect Response) from OmniSQL-32B in Case Study 1**

**Question:** Female refers to SEX = 'F'; born in 1964 refers to YEAR(Birthday) = 1964; admitted to the hospital refers to Admission = '+'. How many female patients born in 1964 were admitted to the hospital? List them by ID.

**Golden SQL Query:**
```sql
SELECT ID FROM Patient WHERE STRFTIME('%Y', Birthday) = '1964' AND SEX = 'F'
AND Admission = '+'
```

**Assistant:**
Let me solve this step by step.
<think>
1. **Identify the goal**: Count how many patients satisfy all conditions and list their IDs. Conditions: 'SEX = 'F'', birth year = 1964, and 'Admission = '+''.

2. **Relevant table and columns**: Use the 'Patient' table with columns 'ID', 'SEX', 'Birthday', and 'Admission'. Extract year via SQLite's 'STRFTIME('%Y', Birthday)'.

3. **Formulate filters**: Apply a 'WHERE' clause combining the three predicates with 'AND'.

4. **Output**: Listing IDs answers the "List them by ID". A separate count (or a window count) answers "How many".
</think>

<answer>
Here is a correct SQL query matching the requirement to list IDs (Golden SQL):

```sql
SELECT ID
FROM Patient
WHERE STRFTIME('%Y', Birthday) = '1964'
AND SEX = 'F'
AND Admission = '+';
```

If you also want the total count together with each ID in one result set (SQLite):

```sql
SELECT
ID,
COUNT(*) OVER() AS FemalePatientCount
FROM Patient
WHERE STRFTIME('%Y', Birthday) = '1964'
AND SEX = 'F'
AND Admission = '+';
```

**Explanation**:
- 'STRFTIME('%Y', Birthday) = '1964'' extracts year-of-birth.
- 'SEX = 'F'' restricts to female patients.
- 'Admission = '+'' keeps only admitted patients.
- The first query lists all matching IDs; the windowed variant also returns the total number.
</answer>

Figure F.1: Example IV (Incorrect Response) from OmniSQL-32B in Case Study 1

1296
1297
1298
1299
1300
1301
1302
1303
1304
1305
1306
1307
1308
1309
1310
1311
1312
1313
1314
1315
1316
1317
1318
1319
1320
1321
1322
1323
1324
1325
1326
1327
1328
1329
1330
1331
1332
1333
1334
1335
1336
1337
1338
1339
1340
1341
1342
1343
1344
1345
1346
1347
1348
1349

---

**Example V (Correct Response) from XYZ-Text2SQL-R1-32B in Case Study 1**

**Question:** Female refers to SEX = 'F'; born in 1964 refers to YEAR(Birthday) = 1964; admitted to the hospital refers to Admission = '+'. How many female patients born in 1964 were admitted to the hospital? List them by ID.

**Golden SQL Query:**
```sql
SELECT ID FROM Patient WHERE STRFTIME('%Y', Birthday) = '1964' AND SEX = 'F'
AND Admission = '+'
```

**Assistant:**
<think>
1. **Understand the Question**: The question asks for the number of female patients ('SEX = 'F'') who were born in 1964 ('YEAR(Birthday) = 1964') and were admitted to the hospital ('Admission = '+''). The result should list these patients by their 'ID'.
2. **Identify the Table**: The 'Patient' table contains all the necessary information to answer the question, including 'SEX', 'Birthday', and 'Admission'.
3. **Filter Conditions**:
- 'SEX = 'F'': This filters for female patients.
- 'YEAR(Birthday) = 1964': This filters for patients born in 1964. SQLite does not have a 'YEAR()' function, but we can use 'strftime('%Y', Birthday)' to extract the year.
- 'Admission = '+'': This filters for patients who were admitted to the hospital.
4. **Select the Required Columns**: The question asks for the 'ID' of the patients.
5. **Count the Patients**: The question also asks for the number of such patients, so we need to use the 'COUNT()' function.
6. **Order by ID**: The result should be listed by 'ID'.
7. **Construct the Query**: Based on the above analysis, we can construct the query.
</think>

<answer>
Here is the correct SQL query:

```sql
SELECT COUNT(ID) AS Total_Female_Patients, ID FROM Patient WHERE SEX = 'F'
AND strftime('%Y', Birthday) = '1964' AND Admission = '+' ORDER BY ID;
```

However, the above query will not work as intended because 'COUNT(ID)' and 'ID' cannot be selected together without using 'GROUP BY'. Since the question asks for both the count and the list of IDs, we need to separate these two requirements. The correct query to list the IDs is:

```sql
SELECT ID FROM Patient WHERE SEX = 'F'
AND strftime('%Y', Birthday) = '1964' AND Admission = '+' ORDER BY ID;
```

To get the count of such patients, run separately:

```sql
SELECT COUNT(*) AS Total_Female_Patients FROM Patient
WHERE SEX = 'F' AND strftime('%Y', Birthday) = '1964' AND Admission = '+';
```

Thus, the final query to satisfy the "list by ID" requirement is the second one.
</answer>

Figure F.2: Example V (Correct Response) from XYZ-Text2SQL-R1-32B in Case Study 1

---

**Example VI (Incorrect Response) from OmniSQL-32B in Case Study 2**

**Question:** Full name of member refers to first_name, last_name. List the full name of the Student_Club members that grew up in Illinois state.

**Golden SQL Query:**
```sql
SELECT T1.first_name, T1.last_name
FROM member AS T1
INNER JOIN zip_code AS T2 ON T1.zip = T2.zip_code
WHERE T2.state = 'Illinois';
```

**Assistant:**
<think>
1. **Understand the Question**: We need the full names ('first_name', 'last_name') of Student_Club members whose zip code corresponds to Illinois.

2. **Identify Relevant Tables**:
- 'member': contains 'first_name', 'last_name', and 'zip'.
- 'zip_code': contains 'zip_code' and 'state'.
- 'major': contains 'major_id' and 'major_name'.

3. **Determine Join Conditions**:
- 'member.zip = zip_code.zip_code'.
- 'member.link_to_major = major.major_id'.

4. **Filter Conditions**:
- 'zip_code.state = 'Illinois''.
- 'major.major_name = 'Student_Club''.

5. **Select Columns**: Output 'first_name' and 'last_name' from the 'member' table.
</think>

<answer>
Here is the correct SQL query:

```sql
SELECT member.first_name, member.last_name
FROM member m
JOIN major mj ON m.link_to_major = mj.major_id
JOIN zip_code z ON m.zip = z.zip_code
WHERE z.state = 'Illinois' AND mj.major_name = 'Student_Club';
```

**Explanation**:
- The 'INNER JOIN' ensures we match members with both their associated major and their zip code.
- The 'WHERE' clause restricts results to members who are in the state of Illinois and belong to the Student_Club.
- Selecting 'first_name' and 'last_name' outputs each qualifying member's full name.
</answer>

---

Figure F.3: Example VI (Incorrect Response) from OmniSQL-32B in Case Study 2

---

**Example VII (Correct Response) from XYZ-Text2SQL-R1-32B in Case Study 2**

**Question:** Full name of member refers to first_name, last_name. List the full name of the Student_Club members that grew up in Illinois state.

**Golden SQL Query:**
```sql
SELECT T1.first_name, T1.last_name
FROM member AS T1
INNER JOIN zip_code AS T2 ON T1.zip = T2.zip_code
WHERE T2.state = 'Illinois';
```

**Assistant:**
<think>
1. **Understand the Question**: The question asks for the full name ('first_name', 'last_name') of members who are associated with the Student_Club and who grew up in Illinois.

2. **Identify Relevant Tables**:
- 'member': contains 'first_name', 'last_name', and 'zip'.
- 'zip_code': contains 'zip_code' and 'state'.

3. **Join Tables**: Use 'member.zip = zip_code.zip_code' to connect the two tables.

4. **Filter Conditions**: Restrict to 'zip_code.state = 'Illinois''.

5. **Select Fields**: Output 'first_name' and 'last_name' from 'member'.
</think>

<answer>
Here is the correct SQL query:

```sql
SELECT DISTINCT m.first_name, m.last_name
FROM member m
JOIN zip_code z ON m.zip = z.zip_code
WHERE z.state = 'Illinois';
```

**Explanation**:
- The 'JOIN' links each member to their state via the 'zip' code.
- The 'WHERE' clause restricts results to Illinois residents.
- 'DISTINCT' ensures duplicate names are not repeated.
</answer>

---

Figure F.4: Example VII (Correct Response) from XYZ-Text2SQL-R1-32B in Case Study 2

