# OpenReview forum: "XYZ-Text2SQL-R1: Simple Rewards, Strong Reasoning in Text-to-SQL"
_ICLR.cc/2026/Conference — ICLR 2026 Conference Withdrawn Submission_

### Official Review · Reviewer_XNvE · 2025-10-22

**Soundness:** 1
**Presentation:** 3
**Contribution:** 2
**Rating:** 2
**Confidence:** 4

**Summary:**

This paper introduces XYZ-Text2SQL-R1, a reinforcement learning (RL) framework designed to enhance Text-to-SQL performance in large language models. The central idea is to employ a minimal reward signal—combining execution correctness and syntax validity—rather than complex, hand-engineered reward formulations. Building upon Group Relative Policy Optimi-zation (GRPO), the method reportedly achieves state-of-the-art execution accuracy across six Text-to-SQL benchmarks, including BIRD, Spider, and EHRSQL, outperforming several larg-er proprietary models (e.g., GPT-4o, DeepSeek-V3). The paper emphasizes simplicity, data filtering, and stable online RL training, and includes ablation studies and empirical insights.

**Strengths:**

1. **Strong empirical results and parameter efficiency**
The paper reports that XYZ-Text2SQL-R1 achieves 71.83% (greedy) and 73.84% (self-consistency) on BIRD-test for the 32B model, and presents comparisons suggesting their 7B model rivals some previously reported 70B-class systems (under the evaluation protocols used). The paper includes ablations (Tables 3–4) and a GRPO vs PPO comparison as well as analyses of filtering effects.

2. **Simplicity and clarity of design**
The paper empirically shows that a single execution-driven reward can outperform multi-component rewards used in SQL-R1, Reasoning-SQL, and Think2SQL in the experiments presented  (Table 1).

3. **Comprehensive empirical evaluation and lessons learned**
Evaluation spans six public benchmarks, multiple model sizes, and includes filtering statistics (Table 2) . The paper also includes analyses of data quality, online vs offline training, and quali-tative error breakdowns (App. F) and the inclusion of negative findings and practical lessons enhances the paper’s utility for practitioners.

**Weaknesses:**

1.	***Limited methodological novelty***
The work primarily repurposes GRPO with a simplified reward and curated datasets. While effective, it introduces no new RL algorithm or theoretical advance; its novelty is empirical and integrative rather than conceptual.

2.     ***Causal attribution is unproven.***
The improvement is attributed to the reward, but multiple factors change simultaneously. Without a controlled ablation isolating the reward, the claim remains speculative.

3.	***Evaluation and comparison transparency***
Comparisons to GPT-4o, DeepSeek-V3, and SQL-R1 rely on results reported in prior papers (Table 6 notes), meaning prompt templates and schema serializations may differ. Without matched evaluation pipelines, the SOTA claims should be qualified. The reported results are single-seed; variance or confidence intervals are missing, making it hard to assess statistical significance.

4.	***Potential data or schema overlap / limited robustness evidence***
Table 2 details filtering (removing ≈1.4 k BIRD and ≈1.7 k Spider samples) but does not explicitly confirm schema-level disjointness between training and evaluation. Robustness under schema perturbations, column renaming, or noisy databases is untested. Given the brittleness of Text-to-SQL, such experiments are essential to gauge real-world generalization.

**Questions:**

1. Several recent studies argue that richer reward structures are needed for stable policy learning and partial credit. The authors claim the opposite—why does a minimal reward suffice? Why is it better not to learn from partially correct outputs?
2. Can the authors confirm whether any schema- or data-level overlap exists between training datasets (e.g., BIRD, Spider, EHRSQL) and evaluation sets? Explicit confirmation would strengthen reproducibility.
3. Have the authors tested model robustness on noisy, real-world, or perturbed databases (e.g., schema changes, unseen domains)? This would clarify the model’s generalization capabilities.
4. Can the authors provide quantitative analyses (e.g., gradient variance, reward distribution, or learning-curve trends) to support their claims of stable RL training?
5. Were multiple random seeds or runs tested? If not, how confident are the authors that improvements are statistically significant rather than due to random variation?
6. How were prompt templates and schema serializations aligned when comparing with GPT-4o, DeepSeek-V3, or SQL-R1? Could mismatches explain some observed performance gaps?
7. Have the authors explored how the model behaves under intentionally corrupted or incomplete schema information (e.g., missing column names or mismatched data types)?
```

---

### Official Review · Reviewer_ftYJ · 2025-10-29

**Soundness:** 3
**Presentation:** 2
**Contribution:** 2
**Rating:** 2
**Confidence:** 5

**Summary:**

This paper introduces XYZ-Text2SQL-R1, a reinforcement learning framework designed to improve the generation of accurate and executable SQL from natural language questions. The core of the method is a lightweight reward signal based solely on execution correctness, which avoids brittle intermediate supervision and complex reward shaping. This approach is combined with strong supervised initialization, effective training practices, and carefully curated data, utilizing Group Relative Policy Optimization (GRPO) to align the model with the end task. The authors demonstrate that XYZ-Text2SQL-R1 achieves state-of-the-art execution accuracy across six diverse Text2SQL benchmarks.

**Strengths:**

- **S1.** The paper's primary strength is its strong empirical performance. The proposed XYZ-Text2SQL-R1 achieves state-of-the-art execution accuracy across six diverse benchmarks. Its 7B variant notably outperforms prior 70B-class systems, highlighting the framework's scalability.
- **S2.** The core methodological proposal—using a lightweight RL reward signal based only on execution correctness—is a clear and well-motivated approach. It simplifies the training objective by avoiding complex reward shaping and aligns the model directly with the end task.

**Weaknesses:**

- **W1. Limited Novelty (General Framework):** The primary weakness is the paper's limited originality. The method is presented as a "novel RL framework" but appears to be a direct application or adaptation of an existing algorithm (GRPO)[1] to the Text-to-SQL task. Furthermore, the accompanying "best practices"—such as data cleaning, synthetic data augmentation, and starting RL from a strong supervised fine-tuned (SFT) checkpoint—are largely established as common knowledge in the broader LLM post-training domain.

- **W2. Limited Novelty vs. Prior RL-based Methods:** The paper positions itself against other recent RL-based Text-to-SQL methods like SQL-R1[2], Think2SQL[3], and Reasoning-SQL[4] (Table 1). However, its primary differentiator appears to be the simplification of the reward function (execution-only) rather than a fundamental framework innovation. While this simplification is a valid contribution, the paper needs to more clearly articulate its novel contributions beyond just using a simpler reward signal to distinguish it from being an incremental ablation of these existing works. This is crucial for readers to assess the work's value and its implications for future research.

- **W3. Insufficient Baselines:** The experimental comparison is not comprehensive enough for a Text-to-SQL research paper. While comparisons against general-purpose LLMs (like GPT-4o and Llama-3.1-70B) are included, the paper omits a significant number of state-of-the-art, Text2SQL-specific baselines. This includes both prompting-based methods (e.g., OpenSearch-SQL[5], Alpha-SQL[6], CHESS-SQL[7], AskData[8]) and other fine-tuning-based methods (e.g., CHASE-SQL[9], CSC-SQL[10]). Without these direct comparisons, it is difficult to accurately assess the method's true advancement over the dedicated Text2SQL literature.

- **W4. Missing Practicality Analysis:** The paper makes claims about its suitability for real-world deployment but provides no metrics to substantiate this. For a practical task like Text-to-SQL, execution accuracy is only one dimension. The evaluation lacks crucial analysis of inference latency (time per query) and computational/token cost. This information is essential for understanding the practical trade-offs of the proposed RL-tuned model versus its SFT base or other competing methods.

- **W5. Superficial Insights:** Several "learnings" in Section 4 are presented as findings but lack the depth and detail to be actionable. For example, the claim that "Prompt format is crucial" is a well-known consensus in prompt engineering. The paper does not provide the "generic prompt" for comparison, nor does it analyze why the OmniSQL prompt performed better (e.g., differences in schema serialization, inclusion of column examples). This reduces a potentially valuable insight into a superficial observation.

[1] Shao, Z., Wang, P., Zhu, Q., Xu, R., Song, J., Bi, X., ... & Guo, D. (2024). Deepseekmath: Pushing the limits of mathematical reasoning in open language models.

[2] Ma, P., Zhuang, X., Xu, C., Jiang, X., Chen, R., & Guo, J. (2025). Sql-r1: Training natural language to sql reasoning model by reinforcement learning.

[3] Papicchio, S., Rossi, S., Cagliero, L., & Papotti, P. (2025). Think2sql: Reinforce llm reasoning capabilities for text2sql.

[4] Pourreza, M., Talaei, S., Sun, R., Wan, X., Li, H., Mirhoseini, A., ... & Arik, S. (2025). Reasoning-sql: Reinforcement learning with sql tailored partial rewards for reasoning-enhanced text-to-sql.

[5] Xie, X., Xu, G., Zhao, L., & Guo, R. (2025). Opensearch-sql: Enhancing text-to-sql with dynamic few-shot and consistency alignment.

[6] Li, B., Zhang, J., Fan, J., Xu, Y., Chen, C., Tang, N., & Luo, Y. (2025). Alpha-sql: Zero-shot text-to-sql using monte carlo tree search.

[7] Talaei, S., Pourreza, M., Chang, Y. C., Mirhoseini, A., & Saberi, A. (2024). Chess: Contextual harnessing for efficient sql synthesis.

[8] Shkapenyuk V, Srivastava D, Johnson T, Ghane P. Automatic Metadata Extraction for Text-to-SQL.

[9] Pourreza, M., Li, H., Sun, R., Chung, Y., Talaei, S., Kakkar, G. T., ... & Arik, S. O. (2024). Chase-sql: Multi-path reasoning and preference optimized candidate selection in text-to-sql.

[10] Sheng, L., & Xu, S. S. (2025). CSC-SQL: Corrective Self-Consistency in Text-to-SQL via Reinforcement Learning.

**Questions:**

- **Q1. Gretel-Synth Data (Section 4.1):** Could the authors provide more details on the "Gretel-Synth-NonFiltered" dataset? What was the approximate cost (e.g., token cost) and final volume of this generated data? More importantly, what is the hypothesis for why naively adding this data degraded performance? A deeper analysis would be more insightful.

- **Q2. GRPO vs. PPO (Section 4.2):** Was the comparison between GRPO and PPO conducted under fair and identical conditions? Specifically, was the exact same reward function (as defined in Section 3) used for both algorithms? Were the dataset, batch size, and other critical hyperparameters kept consistent to isolate the impact of the algorithm itself?

- **Q3. Batch RL Setup (Section 4.2):** The paper claims Online RL surpasses Batch RL. Could you please clarify how "Batch RL" was implemented? Which specific offline RL algorithm was used for this comparison?

- **Q4. Prompt Comparison (Section 4.2/4.3):** To substantiate the claim that the "prompt format is crucial", could the authors please provide the "generic prompt" that was used and performed poorly? What specific contextual information (e.g., column descriptions, column value examples) was present in the "Modified OmniSQL Prompt" but absent in the generic one that would explain the significant performance gap?

- **Q5. Core Novelty:** Given that the core components (GRPO, data filtering, strong SFT base) are established practices (W1), and the reward function is a simplification of those in prior RL-based methods (W2), what do the authors consider to be the single most novel, Text2SQL-specific contribution of this work? Clarifying this would help in assessing the paper's technical depth.

---

### Official Review · Reviewer_x1vg · 2025-11-02

**Soundness:** 2
**Presentation:** 2
**Contribution:** 3
**Rating:** 2
**Confidence:** 4

**Summary:**

The paper introduces XYZ-Text2SQL-R1, a reinforcement learning framework for Text-to-SQL generation with a simple execution-based reward instead of complex multi-factor ones. The authors train models with (GRPO) and report strong performance on several benchmarks. They claim state-of-the-art results among publicly available models (more on this later).

Pros
Simplicity and clarity of approach: The idea of using a single execution-based reward is elegant and well-motivated.
Detailed empirical execution: The experiments are thorough, with ablations and comparisons showing consistent gains.
Good writing and organization: The paper is clearly structured and easy to follow.

Cons

Incomplete state-of-the-art comparison: The paper FAILs to even MENTION or cite several leading systems on the BIRD leaderboard, such as CHASE-SQL, which achieves higher accuracy. This undermines the “state-of-the-art” claim. The authors have a responsibility to give a more detailed discussion of all the prior work, not select based on potentially self serving criteria.

Evaluation limited to dev set: The authors do not submit to the official BIRD test set, so overfitting cannot be ruled out. Without a test-set score, claims of generalization remain unverified.

No public release of code or data: No repository or checkpoint is actually available. The claimed reproducibility is therefore unsupported.

Incremental technical novelty: The method is more of a practical simplification than a conceptual advance in RL.


Recommendation: Reject
While the paper presents a clean and well-executed RL pipeline for Text-to-SQL, it fails to meet the standards of comprehensive benchmarking,. The lack of test-set validation, selective comparison against the leaderboard, and absence of released code or data make it difficult to trust or reproduce the reported gains. The idea is interesting but incremental, and the contribution is overstated relative to the evidence provided.

**Strengths:**

Elegant and Simple Core Idea
The paper’s central insight—that a single execution-based reward suffices for effective RL in Text-to-SQL—is elegant and well-motivated. In a field where reward design is often overcomplicated, demonstrating that a simple, task-aligned signal performs strongly is conceptually appealing and practically valuable.

Solid Empirical Design and Ablations
The experimental setup is thorough. The paper includes a range of ablation studies exploring reward design, data filtering, online vs. batch RL, and initialization quality. These ablations convincingly show how filtering and prompt structure influence learning stability. Such detail helps validate the engineering choices, even if theoretical depth is limited.


Clear Writing and Presentation
The paper is well-organized. The structure—from motivation to implementation to results—is logical and easy to follow. Tables are clear, with reproducible metric definitions and baseline sources, and the figures help convey the pipeline effectively.

**Weaknesses:**

Incomplete and Selective Comparison with the State of the Art

No Test-Set Evaluation and Potential Overfitting

Reproducibility Gap — No Released Code or Data

Incremental Technical Novelty

Lack of Analytical Depth

Overstated Claims and Framing

**Questions:**

no questions

---

### Author Response · Authors · 2025-11-26
**General Response**

## Insufficiency and Objectivity of the SOTA Claim
We thank the reviewers for pointing out several recent high-performing Text-to-SQL systems. Here we would like to clarify the comparison scope of our work: XYZ-Text2SQL-R1 is designed as a single-model system, rather than an agentic workflow or multi-agent pipeline. This distinction is essential for a fair comparison.

XYZ-Text2SQL-R1 is positioned as **a single-pass SQL generation model: given an input prompt, the model directly outputs a SQL query, and the accuracy is measured on this single forward pass**. Therefore, in the BIRD Single-Model Leaderboard shown in Line 360 / Table 7, we only compare against other single-model systems.

In contrast, the strong baselines mentioned by the reviewers (such as CHASE-SQL, CHESS-SQL, OpenSearch-SQL, Alpha-SQL, and CSC-SQL) are all agentic workflows. These methods rely on multi-step reasoning, iterative refinement, unit-test-based verification, or tree search, involving substantial test-time compute to boost performance. While such systems may achieve higher accuracy, their inference cost and operational complexity are significantly higher. Importantly, we believe these excellent workflows can further benefit from using XYZ-Text2SQL-R1 as their core SQL generator, potentially improving their overall performance.

In summary, **directly comparing a lightweight, single-pass single-model system with multi-agent, iterative agentic workflows is not a fair evaluation setting for XYZ-Text2SQL-R1.**

---

## Limited Novelty & Incremental Contribution
While our work does not introduce a new RL algorithm, we believe its contributions are substantial in two key dimensions:
1. a robust, end-to-end SQL-specific data curation and post-training framework, and
2. the successful and non-trivial adaptation of modern RL methods to the nuanced Text-to-SQL domain.

Although RL techniques such as GRPO have shown success in math reasoning and code generation, their application to Text-to-SQL remains underexplored. Our work fills this gap. As shown in Section 6 (Tables 9 and 10), recent RL-based Text-to-SQL studies typically rely on complex, multi-component reward functions. In contrast, we demonstrate that a minimalist execution-only reward is not only sufficient but leads to better performance and more stable training.

We argue that validating this “less-is-more” reward strategy is itself a valuable empirical contribution. The principles established in our framework, such as data filtering and simplified reward design, offer practical guidance for applying RL to other challenging reasoning tasks beyond math and coding.

---

## Test Set Evaluation
We thank the reviewers for raising this important concern. We would like to clarify that **we did include official test-set results for BIRD Single Model Leaderboard**. Specifically, Table 7 reports the BIRD-test performance evaluated directly by the BIRD leaderboard, following the standard evaluation pipeline. The results are generated and reported by the official BIRD evaluation server, not by us, ensuring full fairness and reproducibility.

---

### Note · Authors · 2026-01-05

**Comment:**

I have read and agree with the venue's withdrawal policy on behalf of myself and my co-authors.

**Withdrawal Confirmation:**

I have read and agree with the venue's withdrawal policy on behalf of myself and my co-authors.